# The Fascinating World of Low-Dimensional Quantum Spin Systems: Ab Initio Modeling

**DOI:** 10.3390/molecules26061522

**Published:** 2021-03-10

**Authors:** Tanusri Saha-Dasgupta

**Affiliations:** S.N. Bose National Centre for Basic Sciences, JD Block, Sector III, Salt Lake, Kolkata 700106, India; tanusri@bose.res.in; Tel.: +91-33-2335-5707

**Keywords:** magnetism, density functional theory, spin Hamiltonian, quantum Monte Carlo

## Abstract

In recent times, ab initio density functional theory has emerged as a powerful tool for making the connection between models and materials. Insulating transition metal oxides with a small spin forms a fascinating class of strongly correlated systems that exhibit spin-gap states, spin–charge separation, quantum criticality, superconductivity, etc. The coupling between spin, charge, and orbital degrees of freedom makes the chemical insights equally important to the strong correlation effects. In this review, we establish the usefulness of ab initio tools within the framework of the N-th order muffin orbital (NMTO)-downfolding technique in the identification of a spin model of insulating oxides with small spins. The applicability of the method has been demonstrated by drawing on examples from a large number of cases from the cuprate, vanadate, and nickelate families. The method was found to be efficient in terms of the characterization of underlying spin models that account for the measured magnetic data and provide predictions for future experiments.

## 1. Introduction

Compounds that have a dimensionality of less than three dimensions have long caught the attention of researchers due to their unconventional properties. Reductions in dimensionality can be structural, as in the case of two-dimensional compounds, such as graphene [1] and metal dichalcogenides [2], or as in nanoclusters [3] and nanowires [4]. Reductions in dimensionality can be also electronic, which may occur in otherwise structurally three-dimensional compounds due to interplay between their geometry and the directional nature of the chemical bonding. Magnetic systems of low dimensionality arise when the anisotropic electronic interaction translates into anisotropic magnetic interaction, thereby reducing the effective dimensionality of the magnetic system.

The history of low-dimensional magnetism begins with the Ising model [5], which considers an infinite chain of spins with nearest-neighbor interactions between preferred components of the spin S.$$H_{Ising}=J\sum_{n}S_{n}^{z}S_{n+1}^{z}$$

The other limit of the Ising model is the isotropic Heisenberg model [6],
$$H_{Heisen}=J\sum_{n}\left(S_{n}^{x}S_{n+1}^{x}+S_{n}^{y}S_{n+1}^{y}+S_{n}^{z}S_{n+1}^{z}\right).$$

The ground states of one-dimensional (1-d) uniform chains of S = 1/2 spins are strikingly different in these two models. While the Ising model leads to an ordered ground state, the ground state is disordered even at T = 0 K in the Heisenberg model. The Onsager’s famous solution [7] of the Ising model extended to a two-dimensional (2-d) square lattice showed the existence of a long-range order at a finite temperature with a magnetic transition temperature comparable to the value of the exchange interaction *J*. On the other hand, the two-dimensional Heisenberg system remains disordered at T ≠ 0, but is ordered at T = 0 K. The low-dimensional magnetism in the isotropic Heisenberg model is given by the Mermin–Wagner theorem [8], which states that a 1-d or 2-d array of spins that is described by an underlying isotropic Heisenberg Hamiltonian cannot show a transition to a magnetically ordered state above absolute zero temperature, with the long-range order being destroyed by any finite thermal fluctuation.

A model that is distinct from those of Ising and Heisenberg arises when the magnetic moments lie perpendicularly to the chosen axis, giving rise to what is known as the XY model [9],
$$H_{XY}=J\sum_{n}\left(S_{n}^{x}S_{n+1}^{x}+S_{n}^{y}S_{n+1}^{y}\right).$$

This model shows a unique form of a long-range topological order formed by the bound pairs of vortices below a certain temperature, which is known as the Berezinskii–Kosterlitz–Thouless (BKT) transition temperature [10].

An important distinction between half-integer and integer spin was put forward by Haldane [11]. S = 1 Heisenberg antiferromagnetic (AF) chains, also known as Haldane chains, are conjectured to have spin singlet ground states with an energy gap between the singlet and triplet excited states, which is in marked contrast with the S = 1/2 Heisenberg antiferromagnetic chains, which show a gapless continuum of spinon with the algebraic decay of spin–spin correlation.

Thus, the magnetic behavior crucially depends on the symmetry (discrete for the Ising model, continuous abelian for the XY model, and continuous non-abelian for the Heisenberg model) of the spin Hamiltonian and dimensionality. It came as a further surprise that the crossover between one and two dimensions of S = 1/2 magnets was not found to be at all smooth. In a sense, spin ladders [12] that are formed by spin chains that are put next to each other show unconventional behavior. While ladders with odd numbers of legs display properties similar to those of spin chains and have a power-law decay of their spin–spin correlations, ladders with even numbers of legs have a spin-liquid ground state with an exponential decay of their spin–spin correlations.

Interestingly, the above-described exotic phenomena happen in magnets with small spins—either S = 1/2 or S = 1—and not in large classical moment systems, as they are governed by the quantum nature of the spins. The importance of quantum fluctuation can be appreciated by considering a two-site problem connected by an antiferromagnetic Heisenberg interaction. The bond energy is minimized if the two spins form a singlet in which $|S_{i}+S_{j}|$ = 0. The classical antiparallel alignment gains energy only from the *z*–*z* part of the Heisenberg interaction, while the fluctuation of the *z*-component also allows for energy to be gained from the spin–flip part, with the quantum correction to classical *z*–*z* energy given by $(-JS/\left(-JS^{2}\right)$ = 1/S [13]. The quantum fluctuation is therefore expected to be strongest for small spins, particularly for S = 1/2 or S = 1.

The realization of abstract low-dimensional quantum spin models for real systems started in the period of the 1970s and 1980s, when real materials whose magnetic behavior resembled that predicted in models were synthesized [14]. The presence of strong quantum fluctuations in the low-dimensional magnetic subsystems of high-T${}_{C}$ superconductors [15] triggered renewed interest in both theoretical and experimental studies of low-dimensional quantum spin systems. The original motivation was guided by the possible connection between the spin gap and superconductivity, an issue that is yet to be settled. However, with the extensive investigation of spin ladders, it became clear that even purely insulating low-dimensional quantum magnets can exhibit very rich phenomena that deserve attention in their own right. With the aid of advanced computational methods, theorists were able to solve a variety of more complex low-dimensional quantum spin lattices—examples include the Shastry–Shutherland model [16], the Kagomé lattice model [17], the Kitaev [18] honeycomb model, etc., and many of them have yet to be found in the real world.

An understanding of this complex world requires close collaboration between chemists and experimental and theoretical physicists. Recently, the huge importance of quantum-mechanical calculations based on ab initio electronic structure calculations has been realized; it can become crucial for the identification of the underlying spin model corresponding to a studied material. The measured magnetic susceptibility is often fitted with assumed magnetic models. This procedure may give rise to non-unique answers due to the rather insensitive nature of the magnetic susceptibility with respect to the details of magnetic models, which is complicated by the effect of inter-chain/inter-layer coupling, crystal fields, spin anisotropy, dilution, and other effects that are present in real compounds. Microscopic understanding is thus required for the sake of uniqueness.

One prominent candidate for the realization of low-dimensional quantum spin models is inorganic transition-metal compounds. Although the exchange integrals for these systems are typically several hundred degrees Kelvin, their specific geometry can reduce them substantially in the spirit of the Goodenough–Kanamori–Anderson rules [19,20], making the low-temperature properties easily accessible. They are better choices than organic systems, for which, in most cases, it is impossible to grow large single crystals, and the exchange paths involving organic ligands can be very complex.

Within the limited scope of the present review, we discuss the theoretical attempts within the framework of ab initio density functional theory coupled with the solution of a model Hamiltonian derived based on ab initio inputs, which is applied to understanding and predictions of low-dimensional quantum spin compounds. Specifically, we will discuss the applicability of an ab initio tool of a muffin-tin orbital (MTO)-based method—namely, N-th-order muffin orbital (NMTO)-downfolding—for this purpose. The review will draw examples from cuprates, vanadates, and nickelates, which are listed in Table 1.

It is worth mentioning at this point that other attempts at ab initio modeling of low-dimensional quantum spin systems also exist, which use a variety of methods, such as the extended Huckel tight-binding method (EHTB) [21], tight-binding fitting of the density function theory (DFT) band structure in terms of Slater–Koster parametrization, and total energy calculations. For representative references, see [22,23,24,25,26]. In addition to quantum Monte Carlo (QMC) and exact diagonalization, methods like the density matrix renormalization group (DMRG) [27], bond-operator theory, variational Ansätze, etc. have also been used for solving the spin Hamiltonian.

## 2. Theoretical Framework

The presence of transition metal ions in quantum spin systems makes the electron–electron correlations in their unfilled *d*-shell a dominant effect. Together with the strong correlation effect, the true nature of a magnetic exchange network is often found to not be what is expected from the crystal structure. The theoretical framework must thus include the structural and chemical details. Microscopic investigations demand the involvement of both ab initio methods and many-body effects.

Starting from a Hubbard model [28] description, which describes the competition between the kinetic energy—governed by the hopping interaction, $t_{ij}$—and electron–electron correlation—governed by the Hubbard onsite interaction, *U*—and integrating out the double occupancy in the strong correlation limit (U/t≫ 1), the *t*-*J* model [29] is obtained (where *J* is the magnetic exchange). For half-filling, the *t*-*J* model gives rise to the relevant spin Hamiltonian for studying the quantum spin system. To add chemical reality to such physicists’ models, density function theory (DFT) [30] calculations are carried out with the choice of an exchange-correlation functional of the local density approximation (LDA) [31] or generalized gradient approximation (GGA) [32]. In order to represent only the degrees of freedom associated with magnetic ions, in terms of construction of an effective low-energy Hamiltonian, a highly successful approach has been the downfolding technique within the framework of the *n*-th-order muffin-tin orbital (NMTO) method [33], which relies on the self-consistent DFT potential borrowed from linear muffin-tin orbital (LMTO) [34] calculations.

Within the NMTO method [33], the basis sets may be chosen to span selected energy bands with as few basis orbitals as there are bands by using the downfolding method of integrating degrees of freedom that are not of interest. The method can be used for direct generation of Wannier and Wannier-like functions. This leads to a deterministic scheme for deriving a low-energy model Hamiltonian starting from a complicated DFT band structure. As it is free from fitting parameters, this scheme takes into account the proper renormalization effect from degrees of freedom that are integrated out, and thus retains the information of wave-functions and captures the correct material dependence.

Figure 1 shows the application of NMTO-downfolding to construct the low-energy Hamiltonian for V${}_{2}$O${}_{3}$ [35] and high-T${}_{c}$ cuprate, HgBa${}_{2}$CuO${}_{4}$ [36]. In the case of the former, the bands around the Fermi level (set as zero energy in the figures) are spanned by V $t_{2g}$ states, while for the latter, they are spanned by the antibonding Cu $x^{2}$−$y^{2}$ state. The blue bands in the top panels are the downfolded bands, which show an almost perfect agreement with the DFT band structure in red within ±1 eV around the Fermi level. The Wannier or Wannier-like functions describing the downfolded bands are shown in the bottom panels, which highlight the pdπ and pdσ antibonding nature of the renormalized V $t_{2g}$ and Cu $x^{2}$−$y^{2}$ functions, respectively, with the head part of the functions shaped as V $t_{2g}$ and Cu $x^{2}$−$y^{2}$ and the tail parts shaped as integrated-out degrees of freedom—predominantly O *p*.

The real space representation of this few-band Hamiltonian facilitates the identification of dominant effective hopping interactions that connect the magnetic centers, which bear information on important exchange pathways. Following this, the magnetic exchanges for the identified exchange paths can be obtained either through use of the super-exchange formula [37] or through calculation of the total energy of the different spin configurations within the LDA + *U* calculations [38] and mapping them onto Heisenberg model.

To calculate the thermodynamic properties of the DFT-derived spin Hamiltonian *H*, in the present review, the stochastic series expansion (SSE) implementation of the quantum Monte Carlo (QMC) [39,40,41] method was primarily used, though in some cases, exact diagonalization was also used.

In the following, a brief description of SSE-QMC is given. For details, see [39]. The thermal expectation value of a quantity A is given by
$$\langle A\rangle =\frac{1}{Z}Tr\left\{Ae^{-\beta H}\right\},Z=Tr\left\{e^{-\beta H}\right\}.$$

Within the stochastic series expansion implementation of the quantum Monte Carlo method, one chooses a basis and performs a Taylor expansion of the exponential operator:$$Z=\sum_{\alpha }\sum_{n=0}^{\infty }\frac{\beta^{n}}{n!}{\langle \alpha |\left(-H\right)}^{n}|\alpha \rangle ,$$
where $H=J\sum_{i,j}S_{i}.S_{j}$. One chooses a standard *z*-component basis, $\left|\alpha \rangle =\right|S_{1}^{z},S_{2}^{z},\ldots ,S_{N}^{z}\rangle$, and performs the summation with Monte Carlo technique. We note that, for the quantum case, *H* consists of non-commuting operators.

For a practical implementation, the DFT-derived magnetic exchanges are used as a starting guess, following which the optimal values of the dominant magnetic exchange, *J*, and the effective *g* factor are obtained by fitting the QMC results for the susceptibility:$$\chi^{th}=\langle {\left(S^{z}-\langle S^{z}\rangle \right)}^{2}\rangle ,$$
where $\mu_{B}$ and $k_{B}$ denote the Bohr magneton and the Boltzmann constant, respectively, and the experimental susceptibility (in [emu/mol]) at intermediate to high temperatures is given via χ = 0.375 $\left(g^{2}/J\right)\chi^{th}$. To simulate the low-temperature region of the susceptibility data, the respective Curie contribution from impurities, such as $\chi^{CW}=C_{imp}/T$, is included.

With the stochastic series expansion implementation of the quantum Monte Carlo method, it is possible to simulate quantum spin models in an external field, examples of which will be given in the following.

## 3. Selected Inorganic Quantum Spin Systems

### 3.1. Cuprates

The cuprate family, with Cu in its 2+ valence state of the $d^{9}$ electronic configuration, which amounts to one hole in the highest occupied *d* state, is perhaps the most studied S = 1/2 quantum spin system family. The discovery of high T${}_{c}$ superconductivity in layered cuprate compounds has raised interest in the role of low dimensionality and the quantum nature of the Cu spins. The synthesis of various cuprates with different possible realizations of coordinations of magnetic sublattices has made this family one of the most popular families in terms of the study of low-dimensional quantum spin systems.

*SrCu*${}_{2}$*O*${}_{3}$—With the parent compounds of cuprate superconductors considered as the example of 2-d lattices of spin 1/2 antiferromagnets, efforts were put forward to understand the crossover from chains to square lattices. However, as mentioned above, the transition from 1-d to 2-d quantum spin systems was found to be highly non-monotonic. Even leg-spin ladders were predicted to possess a spin-liquid ground state, while ladders with odd numbers of legs were predicted to possess properties similar to those of single chains [12].

In this ladder family, the compound Sr${}_{14-x}$Ca${}_{x}$Cu${}_{24}$O${}_{41}$ was experimentally synthesized [42,43] and compared to theoretical predictions of the spin gap and superconductivity. The transport properties were found to be dominated by holes in the ladder planes. The normal state of *x* = 11 was found [44] to show a strong anisotropy between the DC resistivity along and across the ladder direction with $\rho_{⊥}/\rho_{∥}$ ∼ 30 at T = 100 K. The microscopic insight into this observation was obtained in terms of DFT calculations performed for SrCu${}_{2}$O${}_{3}$ [45], a compound that possesses the same kind of Cu${}_{2}$O${}_{3}$ ladder planes as Sr${}_{14-x}$Ca${}_{x}$Cu${}_{24}$O${}_{41}$, as shown in the left panel of Figure 2.

In the first ever application of NMTO-downfolding [33], starting from a full DFT calculation, the low-energy Hamiltonian of SrCu${}_{2}$O${}_{3}$ was constructed in terms of renormalized Cu $x^{2}-y^{2}$ orbitals, which were obtained by integrating out all other degrees of freedom except Cu $x^{2}-y^{2}$. The effective Cu–Cu hopping interactions between Cu sites along the rungs and legs and between ladders were found to be long ranged, as shown in the right panel of Figure 2. This analysis showed effective inter-ladder hoppings to be much smaller than intra-ladder hoppings [45]. Furthermore, intra-ladder hoppings between nearest-neighbor Cu pairs were found to be anisotropic with $t_{∥}\neq t_{⊥}$ [45]. This was explained [45] as a consequence of anisotropic $t_{pd}$ in the chemical Hamiltonian model involving both Cu $x^{2}-y^{2}$ and O *p* degrees of freedom due to effective hopping through paths involving Cu 4 *s* states. Estimates of the conductivity in the model where holes were unbound and confined in the ladder [46] were found to give good agreement with the experiments at temperatures of T >100 K [44].

*CaCuGe*${}_{2}$*O*${}_{6}$—Although the crystal structure of CaCuGe${}_{2}$O${}_{6}$ [47] consists of zig-zagged 1-d chains running along the *c*-axis and alternating between two neighboring bc planes, as shown in the top panel of Figure 3, an experimental study involving magnetization and susceptibility measurements was found to be in disagreement with the magnetic properties of the S = 1/2 Heisenberg chain. Instead, the compound was found to show a spin-singlet ground state with an energy gap of 6 meV [48]. However, unlike the well-known related compound CuGeO${}_{3}$ [49], the spin gap is intrinsic, as no spin-Peierls phase transitions were reported between 4.2 and 300 K. This strongly suggests spin dimer characteristics [50]. A question that the experimental measurements could not answer was that of which Cu pairs constitute antiferromagnetic dimers. NMTO-based downfolding of the ab initio band structure together with solution of the effective spin Hamiltonian for computing the thermodynamic properties was carried out to answer the above question [51]. The DFT-derived low-energy Hamiltonian in an effective Cu $x^{2}-y^{2}$ basis showed [51] that longer-ranged magnetic interactions dominated over the short-ranged interactions, and the third-neighbor Cu–Cu pair was the strongest, followed by the nearest-neighbor (NN) Cu–Cu interaction. This led to a description of systems of interacting dimers, given by: $H_{eff}=J_{3}\sum_{\left(i,j\right)}S_{i}S_{j}+J_{1}\sum_{\left(i,j\right)}S_{i}S_{j}$, where $J_{3}$ and $J_{1}$ are intra- and inter-dimer interactions, respectively. This spin Hamiltonian was solved using SSE-QMC for field-dependent magnetization and magnetic susceptibility, as shown in the bottom panels of Figure 3. The optimal values of $J_{1}/J_{3}$ = −0.2 and $J_{3}$ = 67 K = 5.8 meV were found to provide a good description of both magnetization and susceptibility [48]. The underlying spin model is interesting in its own right, in the sense that in the limit $J_{3}$ = 0, it consists of decoupled gapless $J_{1}$ chains (for both positive and negative $J_{1}$), while it shows a gap in the limit $J_{1}$ = 0. Thus, there should be two quantum-critical points, which were found to be $J_{1}$ ∼ 0.55 $J_{3}$ and $J_{1}$ ∼ −0.9
$J_{3}$, although the parameters for CaCuGe${}_{2}$O${}_{6}$ were far from both the critical points. Microscopic analysis thus established that CaCuGe${}_{2}$O${}_{6}$ can be described as a system of dimers formed by the third NN, s = 1/2 Cu${}^{2+}$, with ferromagnetic one-NN inter-dimer coupling. The edge-shared CuO${}_{6}$ octahedra at NN positions with Cu-O-Cu angles of 92${}^{\circ }$ and 98 ${}^{\circ }$ justify the ferromagnetic nature of $J_{1}$. In contrast to CuGeO${}_{3}$, which is a frustrated $J_{1}$−$J_{2}$ system showing a spin-Peierls phase transition, in the present case, the primary role is played by 3rd NN, with possible frustration arising from the second NN being secondary.

*Cu${}_{2}$Te${}_{2}$O${}_{5}$X${}_{2}$ (X = Cl/Br)—*The Cu${}_{2}$Te${}_{2}$O${}_{5}$X${}_{2}$ compounds were introduced [52] as spin-cluster compounds, where, structurally, the magnetic ions form well-defined clusters (see the left panel of Figure 4), and the crystal is made by periodic repetition of the clusters. At first sight, it appears that the magnetic behavior of the compounds should be dominated by spin clusters with little interaction between them. It thus came as a surprise that these compounds were reported to exhibit long-range magnetic orders with T${}^{N}$ (Br) = 11.4 K and T${}^{N}$ (Cl) = 18.2 K [53]. Modeling of the DFT band structure [54] in terms of effective Cu $x^{2}-y^{2}$ orbitals obtained via application of the NMTO-downfolding procedure showed that the effective hoppings were rather long ranged and involved dominant hopping interactions within the Cu${}_{4}$ cluster as well as between the Cu${}_{4}$ clusters (see the right panel of Figure 4). The authors of [54] highlighted the important role of the halogen X${}_{4}$ ring formed by (X-p)-(X-p) covalent bonding and coupled to respective Cu${}_{4}$ tetrahedrons to mediate the long-ranged inter-cluster interactions, thus establishing the long-range order.

*Na${}_{3}$Cu${}_{2}$Te(Sb)O${}_{6}$—*The low-dimensional magnetic behavior of ordered rocksalt oxides [55,56] Na${}_{3}$Cu${}_{2}$TeO${}_{6}$ and Na${}_{2}$Cu${}_{2}$SbO${}_{6}$, which contain layers of edge-sharing Cu–O and Sb/Te–O octahedra (see the left panel of Figure 5) separated by Na${}^{+}$ ions, has raised some debate. The three most significant in-plane Cu–Cu interactions are the NN interactions, $J_{3}$, edge-shared Cu–Cu interactions, $J_{2}$, and Cu–Cu interactions through intervening Sb/Te octahedra, $J_{1}$ (see the middle panel of Figure 5). The long-ranged interactions $J_{4}$ and $J_{5}$ are expected to be negligible. Interestingly, depending on the signs and relative magnitudes of $J_{1}$, $J_{2}$, and $J_{3}$, the magnetic dimensionality of the compounds can be 0 ($J_{1}$≫$J_{2}$∼$J_{3}$), 1 ($J_{1}$ = $J_{2}$≫$J_{3}$ or $J_{1}$>$J_{2}$≫$J_{3}$), or 2 ($J_{2}$∼$J_{3}$∼$J_{1}$). The measured magnetic susceptibility data (see the right panel of Figure 5) suggested an alternating chain model.

Fitting the susceptibility data with antiferromagnetic–antiferromagnetic (AF-AF) and antiferromagnetic–ferromagnetic (AF-F) models has been tried, and it has been concluded that, based on the fitting criterion, it is very difficult to distinguish between the AF-AF and AF-F models [56]. NMTO-downfolding calculations, as well as total energy calculations, established [57] that $J_{1}$ mediated by the Cu-O-Sb/Te-O-Cu pathway is overwhelmingly the strongest exchange pathway for both materials, and while $J_{3}$ is small, the NMTO calculations predicted [57] both $J_{1}$ and $J_{2}$ to be antiferromagnetic (AF) with $J_{2}$ (Sb) / $J_{1}$(Sb) >$J_{2}$(Te) / $J_{1}$(Te). Comparison of calculated and observed Curie–Weiss temperatures considering the AF-AF and AF-F models showed [57] that the AF-AF model gives significantly better agreement (−75 K/ −56 K calculated value vs. −87 K/−55 K observed value for Sb/Te compounds) compared to the AF-F model (−14 K/−11 K calculated value vs. −87 K/−55 K observed value for Sb/Te compounds). It was thus concluded that the AF-AF alternating chain is the appropriate model for both compounds [57].

*CuTe${}_{2}$O${}_{5}$—*In an attempt to analyze the effect of lone-pair cations, such as Te${}^{4+}$, on the magnetic behavior of Cu${}^{2+}$ systems, the CuTe${}_{2}$O${}_{5}$ compound was synthesized and investigated [58]. The crystal structure of the compound [59] consists of edge-shared Cu octahedra forming Cu–Cu dimers, whose corners are shared with TeO${}_{4}$ to form a three-dimensional lattice of CuTe${}_{2}$O${}_{5}$. The measured magnetic susceptibility of CuTe${}_{2}$O${}_{5}$ shows a maximum at T${}_{max}$ = 56.5 K and an exponential drop below T ∼ 10 K, signaling the opening of a spin gap [58]. Electron spin resonance (ESR) data [58] suggested that structural dimers did not coincide with the magnetic dimers. Fitting to magnetic susceptibility data gave rise to a number of possibilities, including a dimer model, an alternating chain model, and an interacting dimer model, while the extended Hückel analysis suggested [58] an alternating chain model. The constructed Wannier-like function of the effective Cu $x^{2}-y^{2}$ via NMTO-downfolding (see the top left panel of Figure 6) showed [60] that, in addition to the formation of a strong pdσ antibond between Cu- $x^{2}-y^{2}$ and O-$p_{x}$/$p_{y}$, the O-$p_{x}$/$p_{y}$ tails of the Wannier function bend towards the Te atom, which is responsible for enhancing the Cu–Cu interactions between different structural dimers. The strongest Cu–Cu interaction, $J_{4}$, was found to be given by Cu pairs belonging to different structural dimers that were connected by two O-Te-O bridges. Two additional in-plane interactions, $J_{6}$ and $J_{1}$, one of which is the intra-structural dimer interaction ($J_{1}$), were found to be appreciable, giving rise to a 2-d coupled dimer model, as shown in the top right panel of Figure 6. An SSE-QMC calculation [41] of the susceptibility data for the proposed 2-d model was found to be in good agreement with the experimental data (see the bottom left panel of Figure 6). Predictions were made for the temperature dependence of magnetization at different values of an external magnetic field (cf. the bottom right panel of Figure 6), which would help to differentiate between the alternating chain model and the 2-d coupled dimer model. Further experimental investigation is needed to settle this issue.

*Cs${}_{2}$CuAl${}_{4}$O${}_{8}$—*The introduction of alternation of nearest-neighbor magnetic interactions into a uniform-chain S = 1/2 AFM Heisenberg model causes its gap-less spectrum to be gapped [61]. The excitation spectrum of a uniform spin chain with both nearest-neighbor and next-nearest-neighbor interactions also becomes gapped if the next-nearest-neighbor (NNN) interactions exceed a certain fraction of the nearest neighbor interactions [62]. It is curious to ask what happens in the presence of both alternation and NNN interactions. For this, one must identify a compound that shows (i) heavily suppressed inter-chain interaction, excluding the formation of a 3-d order and (ii) significant NNN interactions together with alternation. Cs${}_{2}$CuAl${}_{4}$O${}_{8}$, a recently synthesized Cu${}^{2+}$-based compound with a novel zeolite-like structure [63], appears to be a perfect candidate for this. From the fit of the susceptibility data [63], it appears that this compound belongs of the category of non-uniform spin-chain compounds, and it is hard to infer anything further. NMTO-downfolding-based first-principle calculations, as well as total energy calculations, were carried out to provide a microscopic understanding [64]. This gave rise to two different NN interactions between crystallographically inequivalent Cu sites, Cu1 and Cu2, Cu1–Cu2, and Cu2–Cu2, as well as two NNN interactions between Cu1 and Cu2 and Cu2 and Cu2, as shown in left panel of Figure 7. This gave rise to a magnetic J-J-$J^{\prime }$/$J_{nnn}$-$J_{nnn}^{\prime }$ model, giving rise to the first-ever example of a 1-d spin chain with both alternation and competing NNN interactions. Interestingly, the edge-shared NN interactions with near cancellation of Wannier tails at neighboring Cu sites, as shown in the left panel of Figure 7, turned out to be much smaller than NNN interactions for which the Wannier tails at neighboring Cu sites pointed towards each other. The sign of the alternation parameter turned out to be negative, giving rise to the presence of both ferromagnetic and antiferromagnetic nearest-neighbor exchanges, thereby suggesting a rather rich physical system. The solution of the first-principle-derived spin model through the quantum Monte Carlo technique, as shown in the middle panel of Figure 7, provided a reasonable description of the experimentally measured magnetic susceptibility, which shows the presence of a spin gap of ∼3–4 K. The curious nature of the derived spin model prompted further investigation of the model parameter space through exact diagonalization, which showed the possibility of a quantum phase transition from a gap-full to a gap-less situation by tuning the value of $J_{nnn}$ in the presence of competing *J* and $J^{\prime }$, as shown in the right panel of Figure 7 [64].

### 3.2. Vanadates

As opposed to cuprates with the Cu ion primarily in the 2+ valence, which is Jahn–Teller active, with an unfilled occupancy in the $e_{g}$ manifold, vanadate compounds pose the other limit with an unfilled occupancy of the $t_{2g}$ manifold of V ions. Note that the Jahn–Teller activity of $t_{2g}$ ions is expected to be much less than that of $e_{g}$ ions. Furthermore, possible oxidation of V as 4+ and 5+ leads to interesting phenomena, such as charge disproportion, charge ordering, and their influence on magnetic properties.

*$\alpha^{\prime }$-NaV${}_{2}$O${}_{5}$* and *γ-LiV${}_{2}$O${}_{5}$*—Layered vanadates (AV${}_{2}$O${}_{5}$) [65] form an important family of low-dimensional magnets. While CaV${}_{2}$O${}_{5}$ and MgV${}_{2}$O${}_{5}$, with their divalent A sites, contain V in its 4+ state and behave as spin-1/2 ladders that exhibit spin gaps [66,67], monovalent A cation compounds, such as $\alpha^{\prime }$-NaV${}_{2}$O${}_{5}$, [68] γ-LiV${}_{2}$O${}_{5}$ [69], and CsV${}_{2}$O${}_{5}$ [70], have, on average, V${}^{4.5+}$, with important charge and spin fluctuations. The monovalent A cation gives rise to a quarter-filled V $d_{xy}$ band rather than half-filled. Both NaV${}_{2}$O${}_{5}$ and LiV${}_{2}$O${}_{5}$ crystallize in an orthorhombic space group [71,72], with a layered structure of square VO${}_{5}$ pyramids separated by Li${}^{+}$/Na${}^{+}$ ions between the layers. Within the layers, the VO${}_{5}$ pyramids form zig-zagged chains running along the *y*-axis, with two successive zig-zagged chains linked by corner sharing via a bridging O, as shown in the top left panel of Figure 8. The in-plane lattice structure of V ions with possible interactions can be schematically represented as shown in the bottom left panel of Figure 8. Appropriate choices of parameters can present a zig-zagged Heisenberg chain (J(1) ≫ J(b)), a Heisenberg double chain (J(b) ≫ J(1)), or a ladder with one electron per V(1)-O-V(2) rung. The nature and degree of charge disproportionation have a big role in deciding on these three possibilities.

For NaV${}_{2}$O${}_{5}$, it has been established that below a critical temperature, T${}_{c}$ = 34 K, a charge disproportionation appears, 2 V${}^{4.5+}$→ V${}^{5+}$ + V${}^{4+}$, while above T${}_{c}$, all vanadium ions are equivalent (V${}^{4.5+}$) [73]. The magnetic field effect on T${}_{c}$ establishes a zig-zagged ordering of charge-disporportionated V ions [68]. Interestingly, both below and above the charge-ordering transition, the compound has been reported to be insulating [74]. While the insulating nature of charge-ordered phase has been described properly in terms of LDA+*U* calculations, the description of the insulating state of the charge-disordered phase is challenging. The application of the NMTO-downfolding procedure for the construction of a low-energy Hamiltonian in terms of effective V $d_{xy}$ Wannier functions resulted [75] in strongest rung hopping (t${}_{a}$) for 0.38 eV, followed by leg hopping (t${}_{b}$) for 0.08 eV, and inter-ladder hopping for t${}_{1}$ = 0.03 eV and t${}_{2}$ = 0.02 eV. Counter-intuitively, this procedure also resulted in a large diagonal hopping (t${}_{d}$) for 0.08 eV. This strong diagonal hopping had important implications in the description of the underlying Hubbard model corresponding to a two-leg ladder system, which had to include not only local, onsite Coulomb interaction (*U*), but also inter-site Coulomb interaction (*V*), a combination of in-rung and diagonal inter-rung V parameters [75]. The resulting density of states obtained with a cluster dynamical mean field theory (DMFT) [76] solution of the extended Hubbard model (see the top right panel of Figure 8) of the two-leg ladder system highlighted the crucial importance of the inter-site charge fluctuation captured through the *V* parameter in describing the insulating state of charge-disordered NaV${}_{2}$O${}_{5}$.

Unlike $\alpha^{\prime }$-NaV${}_{2}$O${}_{5}$, γ-LiV${}_{2}$O${}_{5}$ does not show any signatures of phase transition, although the crystal structure contains two inequivalent V ions [71], V(1) and V(2), along two legs of the ladder. Modeling in terms of a low-energy Hamiltonian on the basis of electronically active effective V(1)-$d_{xy}$ and V(2)-$d_{xy}$ orbitals (cf. the bottom right panel of Figure 8) showed [77] the onsite energy of V(1) and V(2) to be ± $\epsilon_{0}$ with $\epsilon_{0}$ = 0.15 eV, and showed the rung hopping, $t_{a}$, connecting V(1) and V(2) to be 0.35 eV [77], which is close to that estimated for NaV${}_{2}$O${}_{5}$.

Diagonalization of the simple two-site V(1)-V(2) model gave
$$\frac{\epsilon_{0}}{t_{a}}=\frac{p(2)-p(1))}{2\sqrt{p(1)p(2)}},$$
which yielded p(1)/p(2) = 2.3 for $\epsilon_{0}/t_{a}$ = 0.15/0.35, where p(1)/p(2) are the occupancy of V(1) and V(2). The non-negligible value of p(2) = 0.3 suggests a non-negligible contribution of V(2) to the magnetic moment on the V(1)-O-V(2) rung, which has an important contribution to the microscopic model. For negligible values of p(2), the microscopic model would be that of a zig-zagged chain with $J∼{\left(t_{1}^{\left(1\right)}\right)}^{2}$, since the estimated $t_{b}^{\left(1\right)}$ (−0.06 eV) is much smaller than $t_{1}^{\left(1\right)}$ (−0.18 eV). Consideration of a non-negligible contribution of V(2) changes the scenario quite a bit, with an effective hopping matrix element in the asymmetric rung state:$$t_{b}^{eff}=p\left(1\right)t_{b}^{\left(1\right)}+p\left(2\right)t_{b}^{\left(2\right)}-2\sqrt{p(1)p(2)}t_{d}.$$

The large value of diagonal hopping (−0.1 eV), which was already pointed out for $\alpha^{\prime }$-NaV${}_{2}$O${}_{5}$, makes the effective $t_{b}^{eff}$ large, highlighting the influence of charge ordering on the nature of magnetic coupling [77].

*CsV${}_{2}$O${}_{5}$–*CsV${}_{2}$O${}_{5}$—Belonging to same group as $\alpha^{\prime }$-NaV${}_{2}$O${}_{5}$ and γ-LiV${}_{2}$O${}_{5}$, provides an opportunity to study the influence of A-site cations on the electronic and magnetic properties. The crystal structure of CsV${}_{2}$O${}_{5}$ is somewhat different from those of α-NaV${}_{2}$O${}_{5}$ and γ-LiV${}_{2}$O${}_{5}$. While α-NaV${}_{2}$O${}_{5}$ and γ-LiV${}_{2}$O${}_{5}$ form a double-chained structure of square VO${}_{5}$ pyramids in an orthorhombic space group, the chains are linked together via common corners to form layers, and the layers in the monoclinic structured CsV${}_{2}$O${}_{5}$ are formed by edge-shared V(1)O${}_{5}$ pyramid pairs connected through V(2)O${}_{4}$ tetrahedra, as shown in the top left panel of Figure 9 [78]. The V ions in the tetrahedral coordination are in the 5+ or $d^{0}$ state, while those in pyramidal environment are in the 4+ or $d^{1}$ state. CsV${}_{2}$O${}_{5}$ thus shares the same monoclinic crystal structure as (VO)${}_{2}$P${}_{2}$O${}_{7}$ (VOPO) [79], and KCuCl${}_{3}$ and TlCuCl${}_{3}$ show alternating spin chain behaviors [80]. The measured susceptibility data have been interpreted in terms of the underlying spin dimer model [70].

The right panel of Figure 9 shows the DFT densities of states (DOSs) [81] projected to V 3*d* states for CsV${}_{2}$O${}_{5}$ (top), γ-LiV${}_{2}$O${}_{5}$ (middle), and α-NaV${}_{2}$O${}_{5}$ (bottom). Noticeably, while $\alpha^{\prime }$-NaV${}_{2}$O${}_{5}$ and γ-LiV${}_{2}$O${}_{5}$ show characteristic quasi-1-d van Hove singularities, the DOSs for CsV${}_{2}$O${}_{5}$ show a more 2-d nature, hinting at appreciable inter-dimer interactions. NMTO-downfolding was used to calculate effective V-V hopping, which revealed [81] that, in addition to intra-dimer interactions, $t_{1}$, there are several non-negligible inter-dimer interactions, $t_{2}$, $t_{3}$, and $t_{5}$ (see the bottom left panel of Figure 9) that are mediated by paths through V${}^{5+}$O${}_{4}$ tetrahedra with $t_{1}$ = 0.117 eV, $t_{2}$ = 0.015 eV, $t_{3}$ = 0.097 eV, and $t_{5}$ = 0.050 eV. The strongest hopping $t_{1}$ is, however, significantly smaller than the strongest hopping for $\alpha^{\prime }$-NaV${}_{2}$O${}_{5}$ (0.38 eV) and γ-LiV${}_{2}$O${}_{5}$ (0.35 eV), which is rationalized by the edge-shared path for CsV${}_{2}$O${}_{5}$ as opposed to the corner-shared path for $\alpha^{\prime }$-NaV${}_{2}$O${}_{5}$ and γ-LiV${}_{2}$O${}_{5}$. The proposed 2-d model showed equally good fit to the measured susceptibility compared to the dimer model, stressing the insensitivity of magnetic susceptibility to the details of the spin model. This indicates the need for further experimental studies, such as ESR, inelastic neutron scattering, and Raman scattering, to settle on an underlying spin model.

*VOSeO${}_{3}$—*Spin-gap systems with moderate values of spin gaps are of general interest, as a gap may be closed by a strong enough field, driving a quantum phase transition [82]. Spin dimer systems with weak inter-dimer interactions appear to be attractive systems for realizing this possibility. The VOSeO${}_{3}$ compound, consisting of edge-shared VO${}_{5}$ pairs, forms a probable candidate belonging to this class [83]. NMTO-downfolding calculation for deriving a V $d_{xy}$-only Hamiltonian showed [84] the intra-dimer hopping ($t_{d}$, see the left panel of Figure 10) to be the strongest, followed by the inter-dimer coupling along the *z*-direction (*t*${}_{2}$), with comparable magnitudes ($t_{d}$ = −0.083 eV, $t_{2}$ = −0.079 eV). Other hopping parameters in the yz plane—$t_{4}$, $t_{1}$, and $t_{3}$—are smaller but non-negligible. Two more parameters in the xz plane, $t_{v}$ and $t_{s}$ (see the right panel of Figure 10) also turned out to be non-negligible. Thus, as opposed to the initial suggestion for a spin dimer system, VOSeO${}_{3}$ turned out to be an alternating spin-chain compound with moderate inter-chain interactions.

*Na${}_{2}$V${}_{3}$O${}_{7}$—**Na${}_{2}$V${}_{3}$O${}_{7}$* is the first reported transition-metal-based nanotubular system [85]. The basic structural units are distorted square VO${}_{5}$ pyramids that share corners and edges to give rise to a 2-d sheet-like structure, which, in turn, folds to provide a tube-like geometry with connected rings of nine V atoms, as shown in the top left panel of Figure 11. The Na atoms sitting inside and outside the tubes provide cohesion to the structure. Since the synthesis of this curious structure, several suggestions have been made for the description of the underlying low-energy spin model, which include nine-leg spin tubes [85], mutually intersecting helical spin chains [86], effective three-leg spin tubes with inter-ring frustration, dimerized vanadium moments [87], etc.

Considering the antiferromagnetic spin-1/2 ladder systems, even leg ladders give rise to a spin-singlet ground state with a spin gap, while odd-leg ladders with open boundary conditions result in free spin along one of the legs, resulting in a gap-less situation. Spin tubes, as applicable for Na${}_{2}$V${}_{3}$O${}_{7}$, can be considered as odd-leg ladder with periodic boundary conditions, which, in addition to the spin degrees of freedom, also possess chirality, as shown schematically in the top right panel of Figure 11, and should exhibit a spin gap [88]. The measured susceptibility shows Curie–Weiss behavior at high and low temperatures, with a reduction in the effective magnetic moment from high to low temperatures, and, importantly, no spin gap [87]. The constructed low-energy model keeping the $d_{xy}$ orbital active at three inequivalent V sites showed [89] that due to the complex geometry, the edge-shared V-V couplings were equally as strong as the corner-shared V-V couplings, which is demonstrated in terms of the overlap of Wannier orbitals at different V pairs in the bottom left panel of Figure 11. Neglecting the inter-ring coupling for a first approximation, which is found to be an order of magnitude smaller than the intra-ring couplings, leads to nine-site rings with partial frustration. The partial frustration arises due to the presence of both NN and NNN interactions, though not all NNN interactions appear due to the complex geometry. Exact diagonalization of the nine-site-ring spin model provides a good description of the experimental susceptibility data down to a temperature of a few K [89]. Importantly, the partially frustrated model, as opposed to the fully frustrated model, was crucial for a proper description of the data, as shown in the bottom right panel of Figure 11.

*Zn${}_{2}$VO(PO${}_{4}$)${}_{2}$—*In an attempt to modulate the nature of the magnetic ground state, spin dilution has been attempted through the depletion of magnetic centers. A prominent example is CaV${}_{4}$O${}_{9}$, which is formed by 1/5 depletion of the two-dimensional antiferromagnetic lattice [90]. With a similar motivation, Zn${}_{2}$VO(PO${}_{4}$)${}_{2}$ was studied; 1/4 of the V sites were replaced by Ti [91]. The crystal structure of the pristine compound, as shown in the top left panel of Figure 12, consists of VO${}_{5}$ pyramids. In the ab plane, NN VO${}_{5}$ pyramids are connected via PO${}_{4}$ tetrahedra, while NNN VO${}_{5}$ pyramids are connected via two ZnO${}_{5}$ units. The ab layers are connected via corner-shared PO${}_{4}$ and ZnO${}_{5}$ units to form the tetragonal symmetry [92] of the 3-d structure. Starting from the pristine crystal structure of Zn${}_{2}$VO(PO${}_{4}$)${}_{2}$, every fourth V atom was substituted by nonmagnetic $d^{0}$ Ti${}^{4+}$ ions to achieve dilution of the S = 1/2 lattice formed by $d^{1}$ V${}^{4+}$ ions, as shown in the top right panel of Figure 12. Ti substitution resulted in three inequivalent V sites; the V atoms had no Ti neighbors, thus retaining four in-plane and two out-of-plane V neighbors. The V1 atoms had in-plane Ti neighbors, with two in-plane and two out-of-plane V atoms. The V2 atoms had both in-plane and out-of-plane Ti neighbors, giving rise to only two V neighbors. The NMTO-downfolding calculation for a low-energy model on the basis of the effective V $d_{xy}$ Wannier function for the pristine compound showed [91] that the compound was best described by a weakly coupled two-dimensional S = 1/2 antiferromagnetic square lattice (cf. Figure 12). The NNN and AF V-V magnetic interaction in the ab plane was found to be 2% of the strongest, and the NN and AF V-V magnetic interaction in the ab plane with ferromagnetic interlayer magnetic interaction had a strength of 3% of the NN interaction, which was in good agreement with the conclusions drawn from a neutron scattering experiment [93]. The NMTO-derived spin model for the Ti-substituted Zn${}_{2}$VO(PO${}_{4}$)${}_{2}$ compound turned out to be a coupled S = 1/2 AFM chain. The missing V sites made the in-plane NN V-V interactions unequal along the *a* and *b* directions; the interlayer coupling was found to be of the dimer type. The computed magnetic susceptibility for the pristine compound showed good agreement with experimental data at H = 10,000 Oe [94]. The calculated susceptibility and magnetization (cf. Figure 12) of the substituted compound and their comparison to those of the pristine compound confirmed [91] the change in the magnetic ground state from a long-ranged ordered phase in the pristine compound to a spin-gapped phase in the Ti-substituted compound. This was corroborated by calculated spin wave spectra [95] (cf. Figure 12).

The theoretical prediction needs to be verified experimentally, as the ordering between the Ti and V atoms assumed in the calculations needs to be ensured, which may be challenging.

### 3.3. Nickelates

As opposed to cuprates and vanadates, which have one hole or one electron in the magnetically active Cu${}^{2+}$ or V${}^{4+}$ ions, nickelates, with Ni in their 2+ valence state or $d^{8}$, serve as examples of S = 1 spins with half-filled Ni $e_{g}$ states. The presence of active $e_{g}$ electrons makes the metal–ligand hybridization stronger compared to $t_{2g}$-based systems, such as vanadates. On other other hand, as Ni is a neighbor to Cu, this serves as an excellent opportunity to look for an alternative to cuprates, resulting in the recent study of nickelate superconductivity [96]. This makes the study of the low-dimensional quantum spin systems of nickelates an important topic.

*NiAs${}_{2}$O${}_{6}$—**NiAs${}_{2}$O${}_{6}$*, a member of the 3*d* homologous series AAs${}_{2}$O${}_{6}$ (A = Mn, Co, Ni) [97], shows antiferromagnetic ordering with T${}_{N}$∼ 30 K. The situation became curious with the synthesis of PdAs${}_{2}$O${}_{6}$ with a measured Curie temperature that was five times greater than that of NiAs${}_{2}$O${}_{6}$, ∼150 K [98]. Note that Ni${}^{2+}$ and Pd${}^{2+}$ are examples of $d^{8}$ ions with S = 1 that belong to the 3*d* and 4*d* transition metal series, respectively. The reported magnetically ordered ground states with reasonably high values for the Neél temperature are surprising, since the Ni/PdO${}_{6}$ octahedra do not share corners, edges, or faces in the AAs${}_{2}$O${}_{6}$ structure, which consists of edge-shared As${}^{5+}$ ions in an octahedral oxygen coordination that forms hexagonal layers; different layers are connected by the Ni/PdO${}_{6}$ octahedra [98] (cf. the top left panel of Figure 13). The magnetically active orbitals of $d^{8}$ ions in the octahedral coordination are two $e_{g}$ orbitals, $x^{2}$−$y^{2}$ and $3z^{2}$−$r^{2}$. The Wannier orbitals for $x^{2}$−$y^{2}$ and $3z^{2}$−$r^{2}$ constructed with the NMTO-downfolding procedure of integrating out everything else other than the A $e_{g}$ orbitals (cf. the top right panel of Figure 13) highlighted the bending of O *p* tails of the functions to bond with the sp characters of the nearest As pairs [99]. This enables long-ranged interactions between A-A pairs, although there is no short-ranged interaction mediated by connected O atoms. The dominant hopping paths, as given by the tight-binding Hamiltonian on the downfolded A-$e_{g}$ basis [99], are shown in the bottom left panel of Figure 13. The third NN interactions were found to dominate over second NN and NN interactions. Accounting for the larger $e_{g}$ band width of the Pd compound, the hopping interactions for PdAs${}_{2}$O${}_{6}$ were found to be about 1.4 times larger than those for PdAs${}_{2}$O${}_{6}$ [99]. With Hubbard *U* parameters [99] estimated as $U_{Ni}$ /$U_{Pd}$ ∼2.4, this led to ratios of magnetic exchanges in two compounds of 4.7, which was in good agreement with the ratios of the experimentally measured Neél temperatures of the two compounds.

Finally, the magnetic susceptibility computed with the SSE-QMC of the derived spin model of the two compounds showed exceedingly good agreement with measured data [99], as shown in the bottom right panel of Figure 13.

*NiRh${}_{2}$O${}_{4}$—*With the goal of understanding symmetry-protected topological spin systems, S = 1 spin models on diamond lattices were proposed as potential candidates [100]. This will be a 3-d analogue of the Haldane chain with a gapless 2-d surface state [101]. The A sublattice of spinel compounds offers the possibility of studying diamond lattice magnetism if a magnetic ion can be put at an A site. Magnetic measurements on A sublattice magnetic ion spinels, such as MnSc${}_{2}$S${}_{4}$ (S = 5/2), CoAl${}_{2}$O${}_{4}$ and CoRh${}_{2}$O${}_{4}$ (S = 3/2), and CuRh${}_{2}$O${}_{4}$ (S = 1/2), revealed an ordered magnetic ground state [102,103]. The report of NiRh${}_{2}$O${}_{4}$ with S = 1 ion on the A site therefore created a lot of excitement [104], as the ground state was reported to be non-magnetic.

To provide a microscopic understanding of the nature of the non-magnetic ground state of NiRh${}_{2}$O${}_{4}$, first-principle calculations together with a model study were carried out [105]. We note that Ni at the A site of a spinel is in a tetrahedral coordination, which results in crystal field splitting of Ni *d* states into high-lying $t_{2}$ and low-lying *e* states. The $d^{8}$ occupation thus leads to partially filled $t_{2}$ bands, allowing for spin–orbit coupling to be active among orbitally active degrees of freedom. DFT calculations within a GGA+*U* formulation gave rise to a half-metallic solution, while the inclusion of SOC was necessary to correctly describe [105] the insulating state of NiRh${}_{2}$O${}_{4}$ as it was observed experimentally (cf. Figure 14). This highlights the importance of SOC in the description of SOC in $d^{8}$ systems in tetragonal coordinations, which, as opposed to octahedral $d^{8}$ systems, have active orbital degrees of freedom. The orbital moment at Ni site calculated with the DFT turned out to be [105] large, ∼1 $\mu_{B}$, supporting the formation of an S = 1, L${}_{eff}$ =1 state. The other important input from DFT [105] was the large Ni–Rh hybridization, triggered by the near degeneracy of Ni $t_{2}$ and Rh $t_{2g}$ states in the down-spin channel, as shown in Figure 14. The substantial mixing between Ni–Rh states, in addition to NN Ni–Ni interactions, gave rise to NNN Ni–Ni interactions, as evidenced in the overlap of tails of Ni $t_{2}$ Wannier functions placed at NNN Ni sites and at intervening Rh sites (see Figure 14). The calculated values of magnetic exchanges gave $J_{1}$∼ 1.2 meV and $J_{2}$ ($J_{2}^{\prime }$, $J_{2}^{\prime \prime }$) ∼0.4 $J_{1}$, suggestive of strong magnetic frustration.

Following the DFT results of the L${}_{eff}$ = 1 and S = 1 state, one can write the single-site Hamiltonian as
$$H=-\delta L_{z}^{2}+\lambda \vec{L}.\vec{S}$$

The DFT results gave δ≫λ, with δ, the spin-averaged tetrahedral crystal field splitting between Ni $d_{xy}$ and Ni $d_{yz}/d_{xz}$∼ 100 meV, and the spin–orbit coupling λ∼ 10 meV. The solution of the spin-site Hamiltonian showed the ground state to be a nonmagnetic singlet, as observed experimentally. However, contrary to expectation of a topological quantum paramagnet [106], it turned out to be a spin–orbit entangled singlet state. Incorporating the inter-site interaction via a $J_{1}$−$J_{2}$ Heisenberg exchange model provided a description of the measured inelastic neutron-scattering results [104].

*Sr${}_{3}$NiPt(Ir)O${}_{6}$—*These compounds belong to K${}_{4}$CdCl${}_{6}$-type structures [107] consisting of (${\mathrm{BB}}^{\prime }$O${}_{6}$)${}^{-6}$ chains formed by alternating BO${}_{6}$ trigonal biprisms and $B^{\prime }$O${}_{6}$ octahedra, where the chains are separated by intervening $A^{2+}$ cations, as shown in the top left panel of Figure 15. The construction of magnetically active Ni *d*${}_{xz/yz}$ Wannier functions [108] highlighted the strong hybridization between Ni and Ir states, which is responsible for the Ni–Ni intra-chain interactions through Ir. For Sr${}_{3}$NiPtO${}_{6}$, the intra-chain Ni–Ni interactions occur between magnetically active half-filled Ni $d_{xz/yz}$ levels through oxygen-mediated superexchange paths, which, in accordance with the Kugel–Khomskii picture, gave rise to antiferromagnetic interactions. On the other hand, the interactions between half-filled Ni $d_{xz/yz}$ and one of the Ir $t_{2g}$ states turned out to be antiferromagnetic. Additionally, there exists a direct exchange between the two, which turned out to be of a ferromagnetic nature. A large antiferromagnetic inter-chain interaction for Sr${}_{3}$NiIrO${}_{6}$ was noticed [108] (cf. Figure 15), presumably explaining the antiferromagnetic couplings observed in experiments [109]. The inclusion of spin–orbit coupling showed [108] magnetocrystallic anisotropy to be an easy axis (chain direction) for Sr${}_{3}$NiPtO${}_{6}$, while it to was perpendicular to the chain direction for Sr${}_{3}$NiIrO${}_{6}$.

*SrNi${}_{2}$V${}_{2}$O${}_{8}$—*SrNi${}_{2}$V${}_{2}$O${}_{8}$ serves as a candidate material for studying the closing of the spin gap by an external magnetic field in an S = 1 Haldane chain compound [110]. While the original conjecture of Haldane is applicable to a strictly one-dimensional chain, the behavior of the real compound is complicated by the presence of inter-chain interaction, single-ion anisotropy, etc. [111], resulting in the need for microscopic investigation. In SrNi${}_{2}$V${}_{2}$O${}_{8}$, edge-sharing NiO${}_{6}$ octahedra form zig-zagged chains that are connected to each other by VO${}_{4}$ tetrahedra, giving rise to three-dimensional connectivity [112]. Low-energy modeling in terms of constructed Ni $e_{g}$ Wannier functions demonstrated the effectiveness of hybridization with V degrees of freedom and generated [113] well-defined exchange paths beyond nearest-neighbor intra-chain Ni–Ni interactions. This gave rise to both longer-ranged intra-chain (NNN) and inter-chain interactions, as shown in the top left and right panels of Figure 16. The derived spin model thus consisted of $J_{1}$, $J_{2}$, and $J_{3}$, and $J_{4}$, $J_{1}$, $J_{2}$ were the nearest- and next-nearest-neighbor intra-chain interactions, respectively, while the latter two were the inter-chain interactions. Overlap plots of the Ni $e_{g}$ Wannier functions confirmed [113] (cf. the middle panels in Figure 16) the exchange paths mediated by the V atoms. The magnetic-field-dependent magnetization (cf. the bottom left panel of Figure 16) calculated through the application of SSE-QMC to the derived spin model showed [113] that the critical magnetic field necessary for closing of the spin gap was markedly different from the estimated value of 0.4 $J_{1}$ when considering a strictly 1-d spin-chain model, and further established the effectiveness of $J_{2}$ in tuning the spin gap. Studies carried out with bi-axial strain [113] showed a monotonic decrease in the spin gap value upon increase in the in-plane lattice constant (cf. the bottom right panel of Figure 16). This was found to be caused by the modulation of the $J_{2}$/$J_{2}$ ratio due to compressive strain and the change in the $J_{1}$ value due to tensile strain. This study predicts that bi-axial strain is an effective tool for tuning the spin gap of this compound, which should be verified experimentally. This also opens up the possible strain-induced closing of the spin gap by driving a quantum phase transition from a gap-full to a gap-less situation.

## 4. Summary and Outlook

Low-dimensional quantum magnetism offers a playground for envisaging highly non-trivial and versatile phenomena. One of the key requirements for studying these systems is the identification of the appropriate spin model for describing a given material. Given the success of density functional theory in describing complex materials, it is a natural choice to apply ab initio DFT methods to the problem of low-dimensional quantum magnetism. The strong correlation effect that dictates the properties of quantum spin systems, however, prohibits the direct usage of the DFT for this purpose. Instead, it is much more pragmatic to filter out the DFT’s output to arrive at a low-energy Hamiltonian that contains only magnetic degrees of freedom. In this review, we advocate for the NMTO-downfolding technique as an intelligible, fast, and accurate DFT tool to be used for the filtering. This procedure, which takes the renormalization using the degrees of freedom associated with non-magnetic ions into account, provides information on the relevant exchange paths that connect two magnetic ions. Armed with the knowledge of relevant exchange paths, the corresponding magnetic exchanges can be obtained by employing a superexchange formula from information on the real-space representation of a low-energy Hamiltonian, or in terms of the LDA + *U* total energy calculations of different spin arrangements. The applicability of the proposed method has been illustrated by considering a variety of low-dimensional quantum spin compounds belonging to the cuprate, vanadate, and nickelate families. Table 2 lists a description of the proposed spin model in each case. Most often, the description of the underlying model turned to be different from what may be anticipated based of the crystal structure. The validity of the models was checked by comparing the computed magnetic susceptibility and magnetization with measured magnetic data. Predictions were also made for future experiments in terms of designing spin gaps or closing spin gaps through the application of an external magnetic field, biaxial strain, etc. The theoretical predictions should motivate future experiments in this exciting area of quantum materials.

Finally, we would like to mention that the above discussion is pertinent for 3-d TM-based spin systems only. For 4-d or 5-d TM-based spin systems, in addition to isotropic Heisenberg terms in the Hamiltonian, the presence of non-negligible spin–orbit coupling may give rise to Kitaev, Dzyaloshinskii–Moriya, and off-diagonal anisotropic terms, as discussed, for example, for α-RuCl3, Na${}_{2}$IrO${}_{3}$, and α-Li${}_{2}$IrO${}_{3}$ [114].

## Figures and Tables

**Figure 1 molecules-26-01522-f001:**
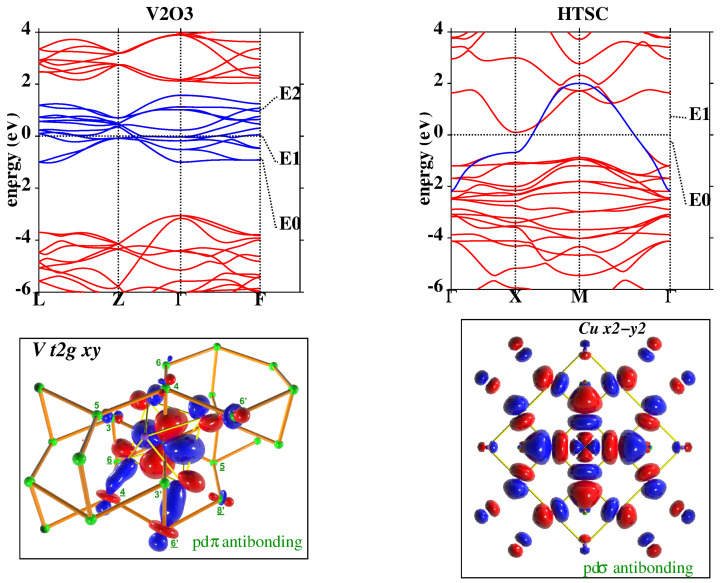
(**Top**) Band structure. of the *n*-th-order muffin-tin orbital (NMTO)-downfolded V $t_{2g}$ bands of V${}_{2}$O${}_{3}$ (**left**) and Cu $x^{2}$−$y^{2}$ bands of Hg cuprate (**right**) in blue lines in comparison to the full density functional theory (DFT) band structure in red lines. E${}_{0}$, E${}_{1}$, etc. denote the energy points used in the energy-selective NMTO-downfolding procedure. (**Bottom**) The NMTO-downfolding-generated Wannier-like functions of V xy in V${}_{2}$O${}_{3}$ (**left**) and Cu $x^{2}$−$y^{2}$ in Hg cuprate (**right**). The central parts of the Wannier functions are shaped according to the active degrees of freedom—namely, V xy or Cu $x^{2}$−$y^{2}$—while the tails of the Wannier functions are shaped according to integrated-out orbitals with significant mixing with active degrees of freedom. The oppositely signed lobes of the Wannier functions are colored in red and blue. This figure is adapted from [35,36].

**Figure 2 molecules-26-01522-f002:**
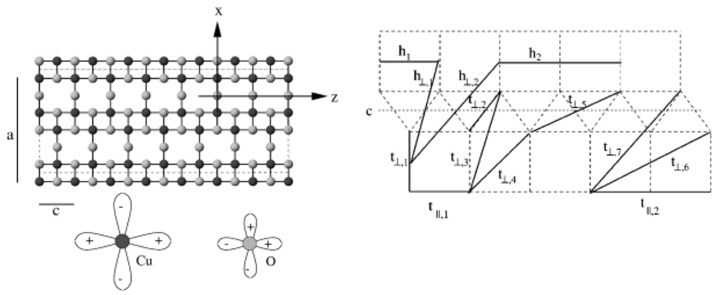
(**Left**) The structure of the ladder compound containing CuO${}_{2}$ planes, with Cu and O atoms marked as black and gray balls, which host Cu $x^{2}$−$y^{2}$ and O $p_{x}/p_{y}$ orbitals. (**Right**) Various effective Cu–Cu intra-ladder hoppings along the rungs and legs, as well as inter-ladder hoppings. Adapted from [45].

**Figure 3 molecules-26-01522-f003:**
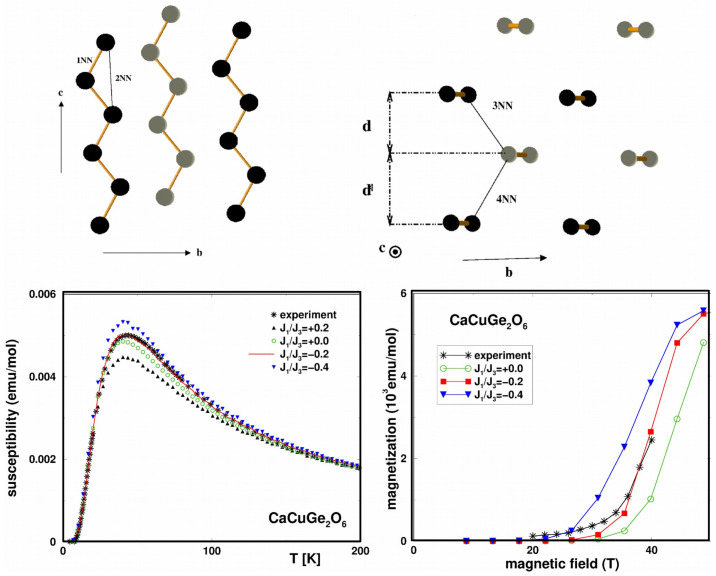
(**Top panels**): Cu-only lattice of CaCuGe${}_{2}$O${}_{6}$ showing zig-zagged chains in the bc planes (**left**) and long-ranged dimers formed in the ab plane (**right**). Atoms belonging to the same and different planes are marked in black and gray colors. (**Bottom panels**): Calculated magnetic susceptibility (**left**) and magnetization (**right**) of the interacting $J_{3}$−$J_{1}$ dimer model in comparison to the experiment [48] for different choices of $J_{1}$/$J_{3}$. Adapted from [51].

**Figure 4 molecules-26-01522-f004:**
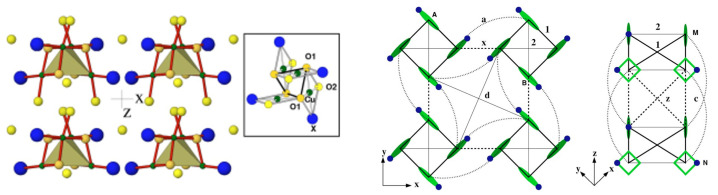
(**Left**) Crystal structure of Cu${}_{2}$Te${}_{2}$O${}_{5}$X${}_{2}$ (X=Cl/Br) with a Cu tetrahedron formed by four Cu atoms in a square environment of oxygen and halogen atoms, which is shown in the inset. Oxygen atoms sitting in the intra-tetrahedral and inter-tetrahedral positions are marked as yellow balls, and halogen atoms, as part of the square environment surrounding Cu${}^{2+}$, are marked as blue balls. (**Right**) Various intra-tetrahedral and inter-tetrahedral Cu–Cu interactions, with filled circles denoting the directions of halogen sites in the square plane surrounding the Cu sites. Adapted from [54].

**Figure 5 molecules-26-01522-f005:**
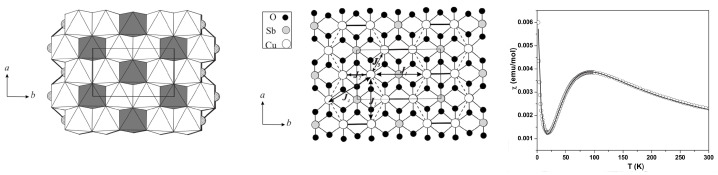
(**Left**) Crystal structure of Na${}_{3}$Cu${}_{2}$SbO${}_{6}$, showing edge-shared Cu octahedra separated by SbO${}_{6}$ octahedra. Na${}_{3}$Cu${}_{2}$TeO${}_{6}$ has a similar structure. (**Middle**) Various Cu–Cu interactions, with the thick line denoting nearest neighbor (NN) Cu–Cu interactions ($J_{2}$), the thin arrowed line marking interactions through Sb(Te)O${}_{6}$ octahedra ($J_{1}$), and the dashed line denoting inter-chain interactions ($J_{3}$. (**Right**) Measured magnetic susceptibility data (circles) fitted with the antiferromagnetic–antiferromagnetic (AF-AF) model (solid line). Adapted from [57].

**Figure 6 molecules-26-01522-f006:**
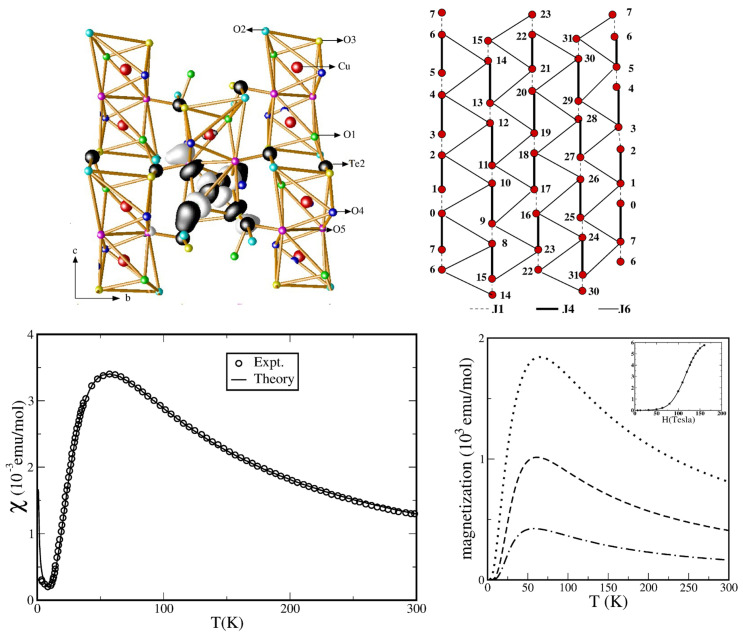
(**Top, left**) Effective Cu $x^{2}$−$y^{2}$ Wannier function of CuTe${}_{2}$O${}_{5}$, as obtained in an NMTO-downfolding calculation, with lobes of opposite signs colored as black and white. Noticeable is the bending of the tail of the Wannier function at the O site to the Te atoms, thus reflecting strong Te-O hybridization. (**Top, right**) The interacting dimer model of CuTe${}_{2}$O${}_{5}$ defined on a 32-site lattice, with dimer interactions marked as thick, solid lines and two inter-dimer interactions as thick, solid, and dashed lines. (**Bottom, left**) Computed magnetic susceptibility in comparison to experimental data [58]. (**Bottom, right**) Computed temperature-dependent magnetization in external magnetic fields of 12.7 T (dash-dotted), 31.7 T (dashed), and 63.4 T (dotted). The inset shows the plot of magnetization as a function of the magnetic field at 10 K. Adapted from [60].

**Figure 7 molecules-26-01522-f007:**
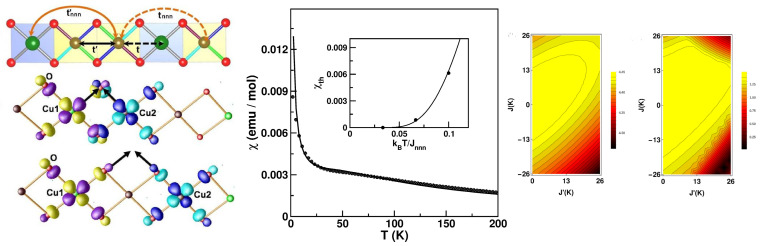
(**Left**) The effective Cu–Cu interactions in the chain compound Cs${}_{2}$CuAl${}_{4}$O${}_{8}$ (**top**), and the overlap of effective Cu Wannier functions placed at NN Cu sites (middle) and next-nearest-neighbor (NNN) Cu sites (bottom ). Positive and negative lobes of wave-functions are colored in yellow/blue and magenta/cyan at site 1 and site 2. (**Middle**) The calculated magnetic susceptibility of the derived spin model, which is shown as a solid line, in comparison with the experimental data, which are shown as symbols, in the presence of a magnetic field of 5 T [63]. The inset shows the susceptibility at H = 0 T in the absence of impurity contributions, which shows the presence of a spin gap. (**Right**) The spin gap in the parameter space of J and $J^{\prime }$ for two choices of $J_{nnn}$ = 78 K (**left**) and = 26 K (**right**). Adapted from [64].

**Figure 8 molecules-26-01522-f008:**
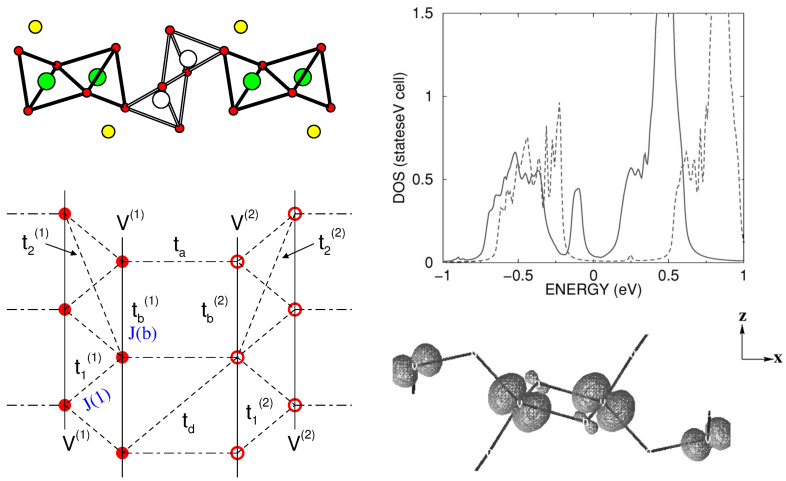
(**Top, left**) The edge-shared and corner-shared VO${}_{5}$ structure of $\alpha^{\prime }$-NaV${}_{2}$O${}_{5}$ and γ-LiV${}_{2}$O${}_{5}$ viewed along the direction of the ladder, with the corner-shared VO${}_{5}$ pyramids forming the rung and edge-shared VO${}_{5}$ pyramids forming the inter-ladder neighbors. The V atoms showing possible charge disproportionation in γ-LiV${}_{2}$O${}_{5}$ are shown as filled green and open white balls, and oxygen atoms are marked as small balls. The Na/Li atoms are shown as yellow balls. (**Top, right**) The density of states of $\alpha^{\prime }$-NaV${}_{2}$O${}_{5}$ with choices of two inter-site Coulomb interactions, *V* = 0.17 eV (solid) and *V* = 0.5 eV, computed with the cluster dynamical mean field theory (DMFT) (**right**). The onsite Coulomb interaction *U* was fixed at 2.8 eV. (**Bottom, left**) Various chain, inter-chain, and rung interactions in the ladder geometry of $\alpha^{\prime }$-NaV${}_{2}$O${}_{5}$ and γ-LiV${}_{2}$O${}_{5}$. (**Bottom, right**) Electronic charge density of low-energy V xy bands of γ-LiV${}_{2}$O${}_{5}$. The bigger and smaller lobe orbitals correspond to partially charge-disporportionated V(1) and V(2) sites. Adapted from [75,77].

**Figure 9 molecules-26-01522-f009:**
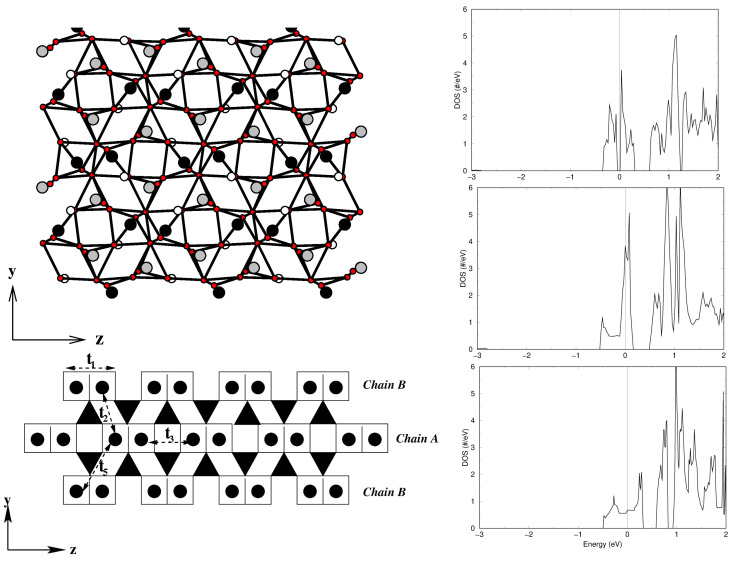
(**Left, top**) The in-plane crystal structure of CsV${}_{2}$O${}_{5}$, with V $d^{1}$ atoms in a pyramidal coordination marked as black balls and V $d^{0}$ atoms in a tetrahedral coordination marked as gray balls. Oxygen atoms are shown as small, red balls, and Cs atoms are not shown. (**Left, bottom**) Various V–V hoppings between edge-shared $d^{1}$ V pairs connected by d${}^{0}$ V in O${}_{4}$ tetrahedra (marked as solid triangles). (**Right**) Comparison of densities of states of V 3*d* states for CsV${}_{2}$O${}_{5}$ (**top**), γ-LiV${}_{2}$O${}_{5}$ (**middle**), and $\alpha^{\prime }$-NaV${}_{2}$O${}_{5}$ (**bottom**). Adapted from [81].

**Figure 10 molecules-26-01522-f010:**
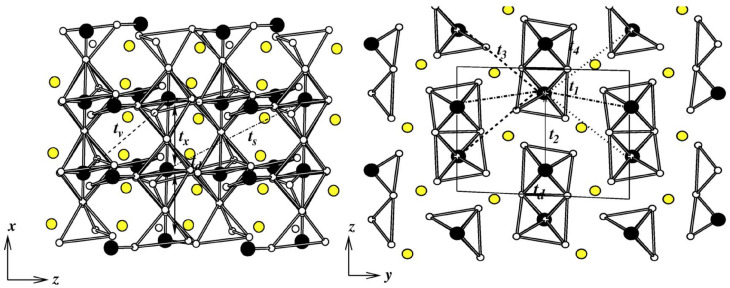
The various out-of-plane (**left**) and in-plane (**right**) inter-dimer V-V interactions in VOSeO${}_{3}$, with V atoms shown as black balls, O atoms as small, white balls, and S atoms as yellow balls. Adapted from [84].

**Figure 11 molecules-26-01522-f011:**
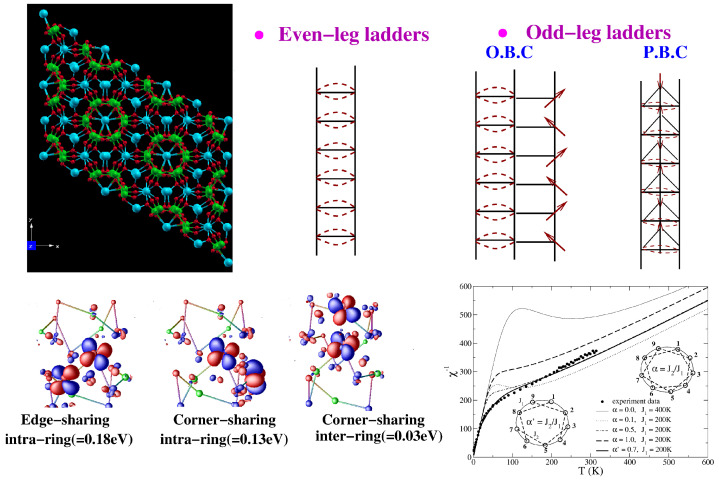
(**Top, left**) The tubular crystal structure of Na${}_{2}$V${}_{3}$O${}_{7}$ with edge- and corner-sharing VO${}_{5}$ pyramids (V atoms in green and O atoms in red), and Na (cyan) atoms occupying the insides and outsides of the tubes. (**Top, right**) Schematic representation of even-leg and odd-leg ladders with open boundary conditions (O.B.C.) and periodic boundary conditions (P.B.C.). (**Bottom, left**) Overlap of effective V xy Wannier functions at various edge-shared and corner-shared positions of V sites in the tube. Shown are the corresponding hopping integrals. (**Bottom, right**) Temperature dependence of the inverse magnetic susceptibility of different spin models compared with the experimental data [87] of the compound. Adapted from [89].

**Figure 12 molecules-26-01522-f012:**
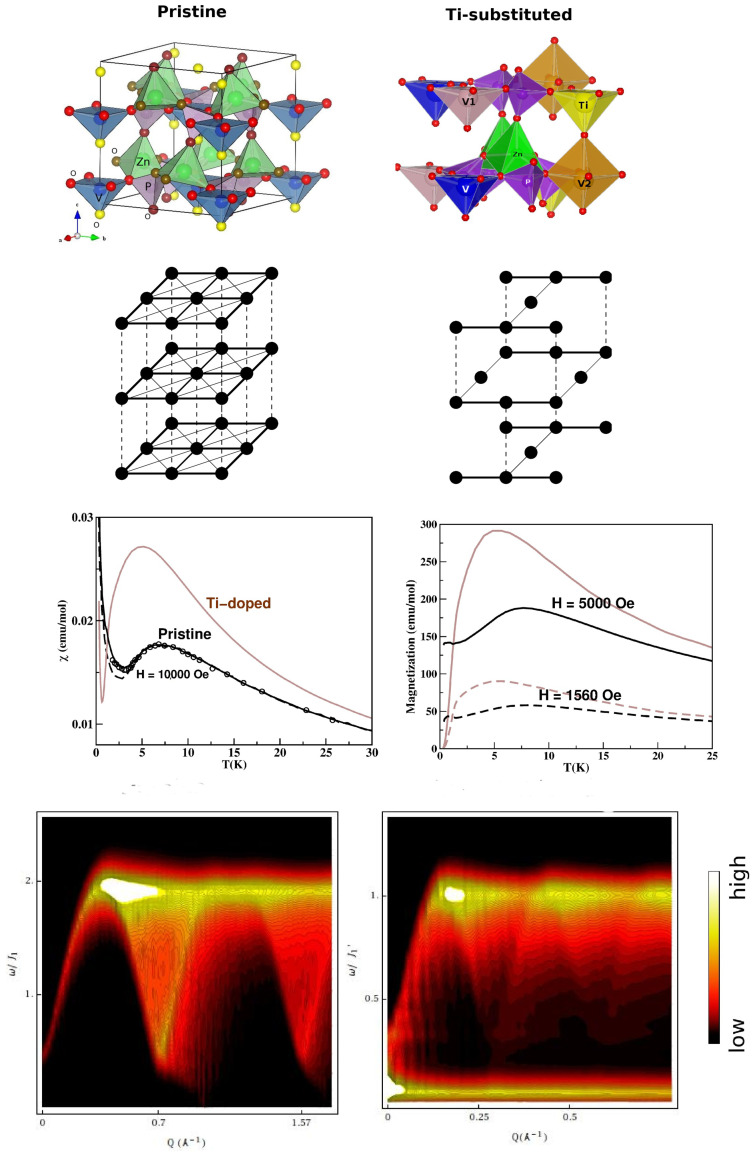
(**Top**) Crystal structures of pristine (**left**) and 1/4 Ti-substituted (**right**) Zn${}_{2}$VO(PO${}_{4}$)${}_{2}$ in a shaded polyhedral representation, with oxygen atoms marked by small, red balls. Second from top: Spin models of pristine (**left**) and 1/4 Ti-substituted (**right**) Zn${}_{2}$VO(PO${}_{4}$)${}_{2}$, with the strongest interactions shown by thick, solid lines, and the two weaker interactions shown by thin, solid, and dashed lines. See the text for details. Third from top: The computed magnetic susceptibility (**left**) and magnetization (**right**) of the pristine (black) and 1/4 Ti-substituted (brown) Zn${}_{2}$VO(PO${}_{4}$)${}_{2}$. For the pristine compound, also shown are the experimental data points for the susceptibility measured at H = 10,000 Oe. (**Bottom**) The computed spin wave spectra for the pristine (**left**) and 1/4 Ti-substituted (**right**) Zn${}_{2}$VO(PO${}_{4}$)${}_{2}$. Figure adapted from [91,95].

**Figure 13 molecules-26-01522-f013:**
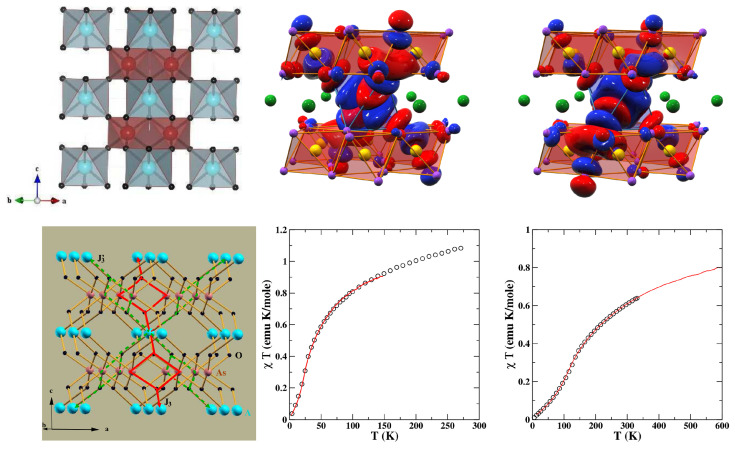
(**Top, left**) Crystal structure of NiAs${}_{2}$O${}_{6}$ with Ni, As, and O atoms shown as blue, brown, and black balls, respectively. PdAs${}_{2}$O${}_{6}$ is isostructural to NiAs${}_{2}$O${}_{6}$. (**Top, right**) The magnetically active Ni/Pd $x^{2}$−$y^{2}$ and $3^{3}$-$r^{2}$ Wannier functions on the downfolded $e_{g}$-only basis. The AsO${}_{6}$ octahedra are marked with two oppositely signed lobes of the wave-functions, which are colored in red and blue. (**Bottom, left**) The spin model involving long-ranged third NN exchange paths, $J_{3}$ and $J_{3}^{\prime }$. (**Bottom, right**) The calculated magnetic susceptibility of NiAs${}_{2}$O${}_{6}$ (left) and PdAs${}_{2}$O${}_{6}$ (right) in comparison to experimental data, which are shown as circles. Adapted from [99].

**Figure 14 molecules-26-01522-f014:**
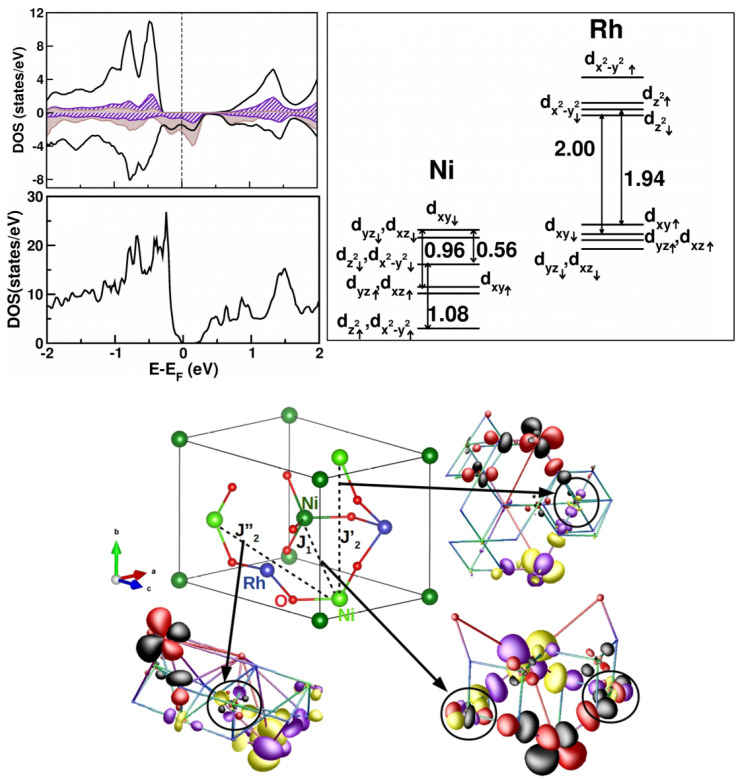
(**Top, left**) The calculated density of states of NiRh${}_{2}$O${}_{4}$ in the GGA + *U* (**top**) and GGA + *U* + SOC schemes of calculations. The zero of energy is set at the respective Fermi energies. For ease of visualization, in the GGA + *U* density of states plot, the states corresponding to the up- and down-spin channels are shown as positive and negative, respectively. Additionally shown are the states projected to Ni *d* (solid, black), Rh *d* (brown shaded), and O *p* (hatched). (**Top, right**) The crystal field splitting at the Ni and Rh sites, as estimated using NMTO calculations on the tight-binding Wannier basis of Ni *d* and Rh *d*. (**Bottom**) The dominant magnetic interactions and the overlap of downfolded Ni Wannier functions at NN and NNN sites. Lobes of opposite signs are colored differently, with the color convention of positive/negative represented by red (yellow)/black (magenta) in sites 1 and 2. Adapted from [105].

**Figure 15 molecules-26-01522-f015:**
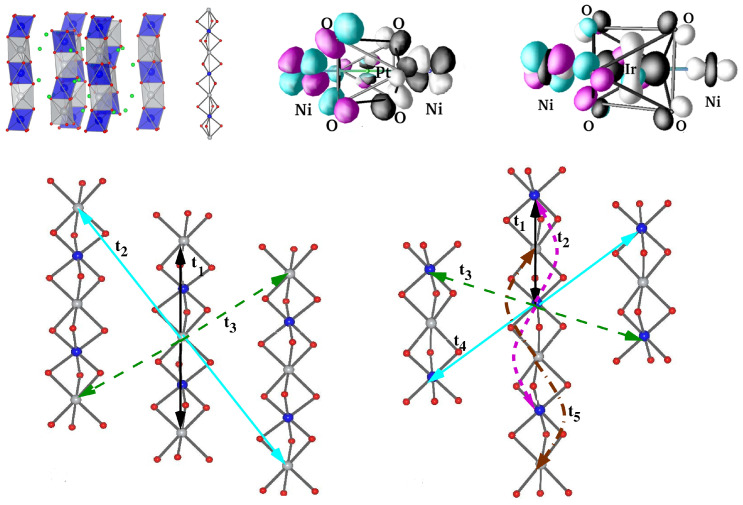
(**Top, left**) The chain structure of Sr${}_{3}$NiPt(Ir)O${}_{6}$ with face-shared NiO${}_{6}$ trigonal biprisms and Ir/PtO${}_{6}$ octahedra that alternate along the chain direction. (**Top, right**) The magnetically active Ni *d* Wannier function for Ni–Pt and Ni–Ir compounds. Note the tail with a weight at neighboring Pt/Ir sites, which is appreciable for the Ni–Ir compound, signaling strong hybridization between the Ni and It *d* states. (**Bottom**) Effective Ni–Ni hopping interactions for Ni–Pt (**left**) and Ni–Ir (**right**) compounds. Adapted from [108].

**Figure 16 molecules-26-01522-f016:**
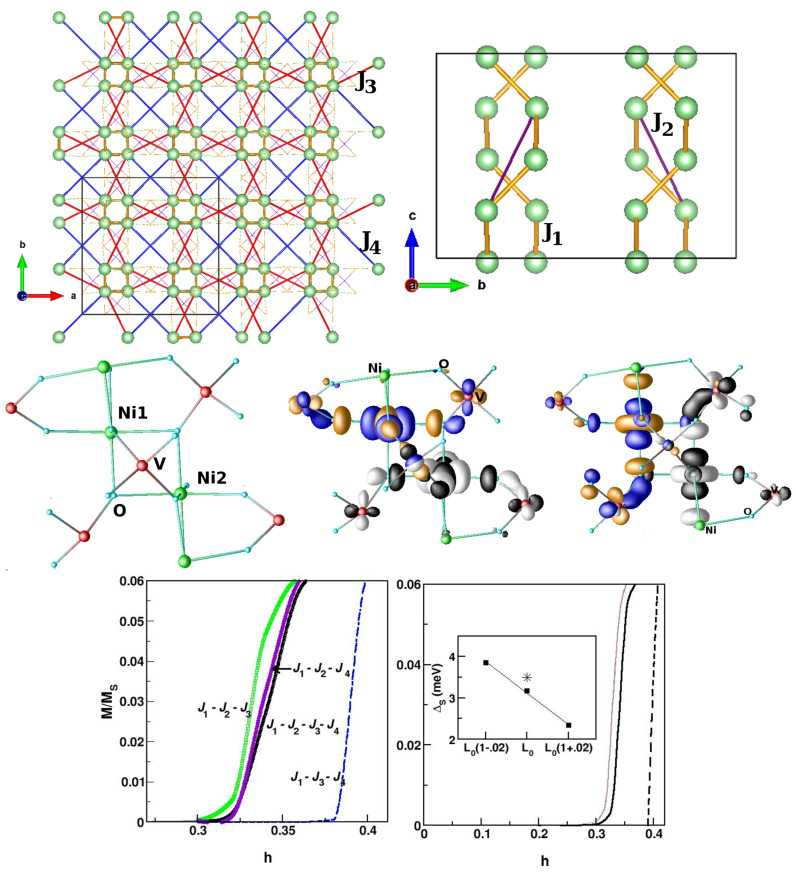
(**Top**) Ni–Ni magnetic interactions in SrNi${}_{2}$V${}_{2}$O${}_{8}$, shown in the plane perpendicular to the chain direction (**left**) and in the plane containing the chain (**right**). Marked are the two dominant intra-chain ($J_{1}$, $J_{2}$) interactions and two inter-chain ($J_{3}$, $J_{4}$) interactions. Middle: Overlap of Ni $x^{2}$−$y^{2}$ and Ni $3z^{2}$−$r^{2}$ Wannier functions at NN positions, connected through edge-sharing oxygens and V atoms. The tails of the Wannier functions at the O positions bend towards the neighboring V atoms due to appreciable V–O hybridization. Bottom: Magnetization calculated as a function of the applied magnetic field for different spin-chain models (**left**) and for unstrained (black, solid line), 2% tensile strained (brown, solid line), and 2% compressive strained (dashed) compounds (**right**). The inset shows the magnitude of the spin gap plotted as a function of the in-plane lattice constant. The experimental estimate of the spin gap for the unstrained compound is marked with an asterisk. Adapted from [113].

**Table 1 molecules-26-01522-t001:** Examples of the quantum spin systems studied in this review.

Compound Name	Magnetic Ion	Occupancy	Spin
SrCu${}_{2}$O${}_{3}$	Cu${}^{2+}$	$d^{9}$	S = 1/2
CaCuGe${}_{2}$O${}_{6}$	Cu${}^{2+}$	$d^{9}$	S = 1/2
Cu${}_{2}$Te${}_{2}$O${}_{5}$X${}_{2}$ (X=Cl/Br)	Cu${}^{2+}$	$d^{9}$	S = 1/2
Na${}_{3}$Cu${}_{2}$Te(Sb)O${}_{6}$	Cu${}^{2+}$	$d^{9}$	S = 1/2
CuTe${}_{2}$O${}_{5}$	Cu${}^{2+}$	$d^{9}$	S = 1/2
Cs${}_{2}$CuAl${}_{4}$O${}_{8}$	Cu${}^{2+}$	$d^{9}$	S = 1/2
γ-LiV${}_{2}$O${}_{5}$	V${}^{4+}$	$d^{1}$	S = 1/2
α-NaV${}_{2}$O${}_{5}$	V${}^{4+}$	$d^{1}$	S = 1/2
CsV${}_{2}$O${}_{5}$	V${}^{4+}$	$d^{1}$	S = 1/2
VOSeO${}_{3}$	V${}^{4+}$	$d^{1}$	S = 1/2
Na${}_{2}$V${}_{3}$O${}_{7}$	V${}^{4+}$	$d^{1}$	S = 1/2
Zn${}_{2}$VO(PO${}_{4}$)${}_{2}$	V${}^{4+}$	$d^{1}$	S = 1/2
Ti-Zn${}_{2}$VO(PO${}_{4}$)${}_{2}$	Ti${}^{4+}$/V${}^{4+}$	$d^{0}$/$d^{1}$	S = 1/2
NiAs${}_{2}$O${}_{6}$	Ni${}^{2+}$	$d^{8}$	S = 1
NiRh${}_{2}$O${}_{4}$	Ni${}^{2+}$	$d^{8}$	S = 1
Sr${}_{3}$NiPt(Ir)O${}_{6}$	Ni${}^{2+}$	$d^{8}$	S = 1
SrNi${}_{2}$V${}_{2}$O${}_{8}$	Ni${}^{2+}$	$d^{8}$	S = 1

**Table 2 molecules-26-01522-t002:** The theoretically predicted spin models of the cuprate, vanadate, and nickelate compounds covered in this review.

Compound Name	Proposed Spin Model	Ref.
SrCu${}_{2}$O${}_{3}$	Two-leg ladder	Ref. [45]
CaCuGe${}_{2}$O${}_{6}$	Coupled dimer	Ref. [51]
Cu${}_{2}$Te${}_{2}$O${}_{5}$X${}_{2}$ (X=Cl/Br)	Interacting spin cluster	Ref. [54]
Na${}_{3}$Cu${}_{2}$Te(Sb)O${}_{6}$	AF-AF chain	Ref. [57]
CuTe${}_{2}$O${}_{5}$	2-d coupled dimer	Ref. [60]
Cs${}_{2}$CuAl${}_{4}$O${}_{8}$	1-d AF-F chain with NN–NNN interaction	Ref. [64]
γ-LiV${}_{2}$O${}_{5}$	Asymmetric spin ladder	Ref. [77]
α-NaV${}_{2}$O${}_{5}$	Spin ladder	Ref. [75]
CsV${}_{2}$O${}_{5}$	2-d coupled spin dimer	Ref. [81]
VOSeO${}_{3}$	Alternating spin chain	Ref. [84]
Na${}_{2}$V${}_{3}$O${}_{7}$	Partially frustrated spin tube	Ref. [89]
Zn${}_{2}$VO(PO${}_{4}$)${}_{2}$	2-d AF spin model	Ref. [91]
Ti-Zn${}_{2}$VO(PO${}_{4}$)${}_{2}$	Coupled AF chain	Ref. [91]
NiAs${}_{2}$O${}_{6}$	2-d spin lattice formed by third-NN interactions	Ref. [99]
NiRh${}_{2}$O${}_{4}$	Spin–orbit entangled singlet	Ref. [105]
Sr${}_{3}$NiPt(Ir)O${}_{6}$	Interacting spin chain	Ref. [108]
SrNi${}_{2}$V${}_{2}$O${}_{8}$	Coupled Haldane chain	Ref. [113]

## Data Availability

Not applicable.

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
