# Peer review of "The Fascinating World of Low-Dimensional Quantum Spin Systems: Ab Initio Modeling"

_molecules, 2021, doi:10.3390/molecules26061522_

Round 1

Reviewer 1 Report

This manuscript is a nice and comprehensive review of inorganic materials
displaying low dimensional quantum spin interactions.

Since the paper is a review of peer reviewed paper, I will comment
on the soundness of results, which I assume to be solid.

Overall the paper is well written and well organized. The selected
systems include examples only of 3d transition metals (actually Cu,
V and Ni) and are representative of a variety of spin Hamiltonians. The
theoretical framework is described adequately. The level of self-citations
(~10%) is acceptable for a review.

Here is a short account of strong and weak points of the paper.

Strong points:
- The message that the magnetic interactions can be substantially
different from the chemical bonding interactions, is well illustrated,
both in the text and in the figures.
- Likewise the message that spin interaction might display a lower
dimensionality than the crystal lattice is well illustrated.

Weak points (to be optionally discussed in the manuscript):
- Only 3d transition metals (TMs) are shown. What about 4d and 5d TMs
where spin orbit interaction and non-collinear spins are important? can
they be described by the same classes of spin Hamiltonian?
- Are there other theoretical methods to calculate the J parameters?
- Are there alternatives to SSE-QMC to simulate the spin Hamiltonians?

line 193: SSH => SSE

I recommend publication of this paper after optional changes. 

Author Response

We are happy to read referee's positive impression about our work, and his/her appreciation. We also thank the referee for pointing out few important shortcomings, as listed below in a point-wise manner.

  1. "Only 3d transition metals (TMs) are shown. What about 4d and 5d TMs
    where spin orbit interaction and non-collinear spins are important? can
    they be described by the same classes of spin Hamiltonian?"

We thank the referee for raising a very important point. We have added a discussion on this along with reference in the concluding section of the manuscript.

2. "- Are there other theoretical methods to calculate the J parameters?"

We have now mentioned of other theoretical methods for J calculations in

the method section.

3. "- Are there alternatives to SSE-QMC to simulate the spin Hamiltonians?"

Alternative methods have been mentioned along with reference.

4. "line 193: SSH => SSE"

We are sorry for the typo which has been corrected now.

Reviewer 2 Report

The presented paper is a review on low-dimensional quantum spin systems. It considers three groups of materials: cuprates, vanadates and nickelates. These kinds of low-dimensional materials are new and exhibit interesting properties. The work is written correctly, both in terms of content and form. I think it is interesting for a wide audience, therefore I recommend its publication in the journal Molecules.

Author Response

We thank the referee for reviewing our manuscript, for his/her encouraging words and for recommending publication of our manuscript.